# Mental Health of Czech University Psychology Students: Negative Mental Health Attitudes, Mental Health Shame and Self-Compassion

**DOI:** 10.3390/healthcare10040676

**Published:** 2022-04-02

**Authors:** Yasuhiro Kotera, Denise Andrzejewski, Jaroslava Dosedlova, Elaina Taylor, Ann-Marie Edwards, Chris Blackmore

**Affiliations:** 1School of Health Sciences, University of Nottingham, Nottingham NG7 2HA, UK; 2Department of Psychology, Middlesex University, Dubai Block 16, Dubai P.O. Box 500697, United Arab Emirates; d.andrzejewski@mdx.ac.ae; 3Department of Psychology, Masaryk University, 602 00 Brno, Czech Republic; dosedlova@mail.muni.cz; 4College of Health, Psychology and Social Care, University of Derby, Derby DE22 1GB, UK; e.taylor3@derby.ac.uk (E.T.); annm.edwards@icloud.com (A.-M.E.); 5School of Health and Related Research, University of Sheffield, Sheffield S1 4DA, UK; c.m.blackmore@sheffield.ac.uk

**Keywords:** self-compassion, Czech university students, mental health, mental health attitudes, mental health shame

## Abstract

High rates of mental health problems are a growing concern in Czech higher education, negatively impacting students’ performance and wellbeing. Despite the serious nature of poor mental health, students often do not seek help because of negative attitudes and shame over mental health problems. Recent mental health research reports self-compassion is strongly associated with better mental health and reduced shame. However, relationships between those constructs remain to be evaluated among Czech students. This study aims to appraise the relationships between mental health problems, negative mental health attitudes, mental health shame, and self-compassion in this population. An opportunity sample of 119 psychology students from a Czech university completed questionnaires regarding these constructs. Correlation, regression, and path analyses were conducted. Mental health problems were positively associated with negative mental health attitudes and shame, and negatively associated with self-compassion. Self-compassion negatively predicted mental health problems, while negative attitudes and shame did not. Last, self-compassion fully mediated the negative attitudes-mental health problems relationship, and partially mediated the shame-mental health problems relationship. Findings suggest self-compassion is essential for mental health in Czech students and associated with negative mental health attitudes and mental health shame. Czech universities can benefit from incorporating self-compassion training into their curricula to protect students’ mental health.

## 1. Introduction

Mental health is high on the national agenda in the Czech Republic. The social and economic transformations that followed the collapse of the socialistic regime took a toll on the mental health of the people of Central and Eastern Europe that persists to this day [1,2,3]. In the Czech Republic, one in five adults are diagnosed with a mental health illness [4]. Alcohol dependence is almost twice as high as in the rest of Europe [5]. The highest prevalence rates of alcohol dependence (16.64%), as well as mood (7.96%) and anxiety disorders (5.42%), have been found in young people aged 18 to 29 years old, predominantly undergraduate university students [4]. The consequences of poor mental health in university students are higher dropout rates and lower academic achievement [6], which is partially mirrored in lower tertiary qualification attainment (i.e., college, university, and vocational courses) in the Czech Republic [7]. Furthermore, Bobak et al. [8] found a high prevalence of depression among a Czech Republic adult sample and were able to establish an inverse relationship between psychological wellbeing and acquired education: well-educated adults in the Czech Republic tended to have poor psychological wellbeing. This trend is present consistently among Eastern European university students [9]. While the government has recently started reforming mental health care, underfinancing and insufficient legislation [10] are still contributing to the substantial treatment gap in the Czech Republic where 83% of people with a mental disorder need care but have not received it [11,12].

Furthermore, mental health illness and its diagnosis in the historical Soviet discourse has mainly served as an instrument of oppression, and led to inhumane and amoral treatment. These conditions resulted in a stigmatization of psychiatry [1] that continues to foster negative attitudes towards the discipline of mental health at large [13]. As attitudes and beliefs about mental health are formed and maintained through cultural knowledge and perceptions, which are often based largely on historical narratives [14], the higher prevalence of mental health stigma in former socialist societies not only poses an additional barrier to close the treatment gap in the Czech Republic [13,15], but can potentially worsen overall mental health in the region [16].

The detrimental effects of negative mental health attitudes (i.e., believing that having a mental health problem indicates that the person is weak and inadequate) are well known. Research suggests that such attitudes impact negatively on self-esteem [17,18], self-efficacy [19], and physical health [20], and that they are a significant obstacle to treating mental health. These socio-cognitive barriers are among the leading obstacles for help-seeking behaviours, followed by mental health knowledge and awareness [21,22,23,24,25,26]. While studies have shown that mental health literacy is associated with more positive attitudes towards mental health [27], only 1% of Czech medical students are genuinely interested in pursuing careers in psychiatry [28]. The extent to which negative mental health attitudes and shame in post-Soviet societies, such as the Czech Republic, affect treatment seeking, has not yet been explored in detail.

### 1.1. Negative Mental Health Attitudes and Shame

It is well established that negative attitudes about mental health can lead to internalisation potentially manifesting in a sense of shame [29,30,31,32,33]. The emotional state of shame is complex and arises when individuals feel that they fall short of internalised socially constructed standards [32,34,35,36]. Shame involves negative self-evaluations and concerns about the judgements of others, and feelings of regret about one’s identity [37,38]. As a marker of psychopathology, shame has been linked to depression [39], anxiety [40], and eating disorders [41]. “Mental health shame” ([42], p. 136)—feeling ashamed for having a mental health problem—is linked in university students to poorer mental health [43] and is especially prominent in students who prepare to enter demanding careers such as business management [44] and health care [45]. Doblytė [13] qualitatively explored feelings of shame regarding mental health problems in an adult Czech sample and observed that shame was a dominant theme for delayed treatment seeking and the adoption of destructive coping strategies to prevent stigmatisation. Though psychopathology is affecting predominately young adults in the Czech Republic [11] and the stigma surrounding mental health in the country remains pervasive [15], the relationship between negative attitudes, mental health shame, and mental health problems has not yet been quantitatively examined in Czech undergraduate students.

### 1.2. Self-Compassion

Research that focused on mental health improvement and shame reduction has consistently identified self-compassion as a protective factor [31,46,47,48,49,50]. Rooted in the tradition of Buddhism [51], self-compassion is related to practicing kindness towards oneself when facing adversity, acknowledging that struggling and suffering is a shared human condition, and becoming mindful and aware of one’s painful thoughts [52]. Self-compassion has been linked to lower rates of depression, anxiety, and stress [52,53,54,55], as well as reduced social comparison [56] and self-criticism [31,47,57,58]. It has also been beneficially linked to life satisfaction, happiness, optimism, and overall wellbeing and better mental health [53,54,57]. Some have investigated the moderating role self-compassion plays in psychopathological symptoms such as rumination and stress [59], and self-criticism and depression [57]. However, the mediating role of self-compassion in the relationship between shame and mental health problems have largely only been explored in the context of eating disorders (e.g., refs. [58,60]). Self-compassion has begun to attract attention in the Czech Republic [61]. Montero-Marin et al. [62] suggested that this may be because practicing self-compassion is influenced by cultural values. Most notably, self-compassion is suggested to be inversely related to indulgence and restraint, as outlined by Hofstede’s [63] Cultural Dimensions Theory. Like many other Eastern European countries, the Czech culture scores comparably lower in the indulgence domain than other Western societies such as Germany, the United Kingdom, and the United States [64]. Restraint and control of impulses and desires is a governing cultural value in the Czech Republic [63]. Therefore, self-compassion might not be a common trait or known skill in Czech culture and thus is a meaningful area to explore to support the national mental health agenda goals of improving the quality of life of people with mental illnesses and widening access to treatment [10].

### 1.3. Mental Health and Emotion Regulation

Disorders of distress such as anxiety and depression have been widely linked to emotional dysregulation [65] and often lead to maladaptive coping behaviours such as substance [66,67] and alcohol abuse [68]. Neurophysiological research suggests that there are three main emotion regulation systems [69], namely the threat, drive, and soothing systems [70]. The threat system functions as an alarm apparatus that elicits feelings of anxiety and anger, resulting in protection-seeking behaviours [70]. The drive system is goal oriented and triggers behaviours that bring pleasure [69]. The soothing system focuses on safety, and reduces distress through nurturing and affection [70]. According to Gilbert [71], taken together these emotion regulation systems, if balanced, form the foundation of mental wellbeing, but cause distress and psychopathology if unbalanced. The societal values of restraint and impulse control in Czech culture [64] could lead to imbalances in some individuals, with greater activation of the threat system, and diminished activation of the soothing system [62]. This in turn could explain lower engagement in self-compassionate behaviours. High activation of the threat system could further explain the high prevalence in mental health illness in the Czech Republic. Accordingly, we theorise that Czech individuals might predominantly operate on the threat system, which could be a plausible explanation for the high prevalence of mental health problems in the country. Therefore, if shame and negative attitudes towards mental health are anchored in the threat system then self-compassion as a soothing mechanism should be able to mediate the negative effects of the overstimulated threat system on mental health.

### 1.4. Aims and Hypotheses

Poor mental health affecting young adults in the Czech Republic can impair educational attainment and quality of life [13]. Recent mental health reform is currently addressing legislative avenues to facilitate accessibility to mental health services and better treatment in the Czech Republic. It is important to consider the cognitive barriers for individuals in accessing support and treatment, which has thus far not been explored in depth.

Given the scarcity of empirical studies investigating this association, the current study aimed to appraise the relationships between mental health problems, negative mental health attitudes, mental health shame, and self-compassion in Czech undergraduate psychology students. We particularly focused on depression, anxiety, and stress as “mental health problems” as these symptoms are most common among the general public and university student populations [72,73]. Additionally, based on Gilbert’s [71] emotion regulatory systems, we investigated if self-compassion would be able to mediate the relationship between (H3) negative mental health attitudes and mental health problems, and (H4) mental health shame and mental health problems among Czech university students.

Our guiding hypotheses for the current study were as follows:

**H1:** 
*Greater mental health problems are significantly associated with (a) negative mental health attitudes, (b) mental health shame, and (c) self-compassion.*


**H2:** 
*Mental health problems are significantly predicted by (a) negative mental health attitudes, (b) mental health shame, and (c) self-compassion.*


**H3:** 
*Self-compassion mediates the relationship between negative mental health attitudes and mental health problems.*


**H4:** 
*Self-compassion mediates the relationship between mental health shame and mental health problems.*


## 2. Materials and Methods

### 2.1. Design

A cross-sectional study design was employed. Ethical approval was granted by the Ethics Panel of the Institute of Psychology, Faculty of Arts, Masaryk University (ID: 035/20).

### 2.2. Participants

Participants were aged 18 years or older and were undergraduate psychology students studying at a Czech university. Students who were on a break from studies were excluded from the study. Students were invited to take part in the study through a class announcement by programme tutors, who were not the authors of this paper, in February 2020. No incentive for participation was given.

An opportunity sample of 130 students agreed to participate. The final sample consisted of 119 students with an overrepresentation of females (*n* = 93; 78%) compared to males (*n* = 20), which is similar to the ratio found in the general psychology population in the Czech Republic [74]. Six students did not disclose their gender. The sample was aged between 19 and 44 (M = 21.87, SD = 3.32 years). The sample satisfied the required sample size of 115 based on statistical a priori power calculations [75]. The majority of students were Czech (*n* = 98; 82%), and the rest were Slovakian students (*n* = 21; 18%). The withdrawn 11 students did not report any reason or complaint.

### 2.3. Measures

Three self-report measures were used in this study (Table 1).

Mental health problems were assessed using the Depression Anxiety and Stress Scale 21 (DASS21), a shorter version of the original DASS42 [76]. DASS21 evaluates the levels of depression (“I found it difficult to work up the initiative to do things.”), anxiety (“I felt I was close to panic.”), and stress (“I felt that I was using a lot of nervous energy.”). Participants reflect on the past week and select how much each statement applies to them on a scale of 0 to 3 (0 = “Did not apply to me at all.”; 3 = “Applied to me very much, or most of the time.”). DASS21 has good reliability: α ≥ 0.87 [77].

Negative mental health attitudes and mental health shame were evaluated using the Attitudes Towards Mental Health Problems (ATMHP), comprising 35 items on a four-point Likert scale (0 = ‘do not agree at all’ to 3 = ‘completely agree’; a higher score indicates more negative attitudes and stronger shame) [30]. This scale consists of four sections: (i) Community Attitudes and Family Attitudes refer to their perception of how their community and family perceive mental health problems, (ii) External Shame appraises their perception of how their community and family would view them if they had a mental health problem, (iii) Internal Shame considers how they would perceive themselves if they had a mental health problem, and (iv) Reflected Shame examines how their family would be perceived if they had a mental health problem (Family-Reflected Shame), and fears of reflected shame on themselves if a close relative had a mental health problem (Self-Reflected Shame). All of the subscales had good internal consistency (α = 0.85–0.97; [30]). Negative mental health attitudes were calculated by totalling the scores for Community and Family Attitudes, and mental health shame was calculated by totalling the scores for External Shame, Internal Shame, and Reflected Shame.

Self-compassion was assessed using the Self-Compassion Scale-Short Form (SCS-SF). This 12-item self-report measure is a shorter form of the 26-item Self-Compassion Scale [52], responded on a five-point Likert scale (e.g., “When something painful happens I try to take a balanced view of the situation.”; 1 = ‘Almost never’ to 5 = ‘Almost always’). The internal consistency was high (α = 0.86) [78].

### 2.4. Data Analysis

Data was screened for outliers and the assumptions of parametric tests. Pearson correlations between mental health problems, negative mental health attitudes, mental health shame, and self-compassion were evaluated using an SPSS version 26 (IBM, Chicago, IL, USA). Multiple regression analyses were performed to identify significant predictors of mental health problems. Path analysis was undertaken to explore effects of negative mental health attitudes in the relationship between mental health shame and mental health problems. Last, mediation analyses were conducted to examine the impact of negative mental health attitudes on the relationship between mental health shame and mental health problems. For the path analyses, the Process Macro 3 for SPSS (IBM, Chicago, IL, USA) [79] was used, with a setting of 5000 bootstrapping re-samples and bias-corrected 95% confidence intervals (CIs).

## 3. Results

Three scores in mental health problems and four scores in self-compassion were identified as outliers using the outlier labelling rule [80], so were winsorised [81]. All variables had good reliability (α = 0.81–0.90; Table 1). Because scores on all variables were not normally distributed (Shapiro-Wilk’s test, *p* < 0.05), they were square-root-transformed to satisfy the assumption of normality.

### 3.1. Correlations (H1)

Pearson’s correlation was conducted to examine the relationship between mental health problems, negative mental health attitudes, mental health shame, and self-compassion. Mental health problems were significantly positively associated with negative mental health attitudes (*r* = 0.26) and mental health shame (*r* = 0.35), and negatively associated with self-compassion (*r* = −0.54). H1 was supported.

### 3.2. Regression (H2)

Multiple regression analyses were conducted to appraise the relative contribution of negative mental health attitudes, mental health shame, and self-compassion to mental health problems (Table 2). At step 1, gender and age were entered to statistically adjust for their effects, and at step 2, negative mental health attitudes, mental health shame, and self-compassion were entered. Adjusted coefficient of determination (Adjusted R^2^) was reported. Multicollinearity was not a concern (the variance inflation factor < 10). After adjusting for demographic information, negative mental health attitudes, mental health shame, and self-compassion accounted for an additional 29% of the variance for mental health problems (*p* < 0.001). Negative mental health attitudes (*p* = 0.23) and mental health shame (*p* = 0.24) were not significant predictors for mental health problems. Self-compassion score was a significant negative predictor of the mental health symptom score (*p* < 0.001). A one-unit decrease in the self-compassion score was significantly associated with a 4.25-unit increase in the mental health problems score.

### 3.3. Mediation of Self-Compassion on Negative Mental Health Attitudes—Mental Health Problems (H3)

Path analysis was conducted, using Model 4 in the Process Macro parallel mediation model [79], to examine whether self-compassion (mediator variable) mediated the relationship between negative mental health attitudes (predictor variable) and mental health problems (outcome variable).

Self-compassion mediated the relationship between negative mental health attitudes and mental health problems (Figure 1). The total effect of negative mental health attitudes on mental health problems, including self-compassion, was significant, *b* = 0.36, *t*(117) = 2.92, *p* = 0.004, CI [0.12, 0.60]. The direct effect of negative mental health attitudes on mental health problems was not significant, *b* = 0.18, *t*(116) = 1.66, *p* = 0.10, CI [−0.04, 0.40]. The indirect effect of negative mental health attitudes on mental health problems, controlling for self-compassion, was significant, *b* = 0.18, CI [0.06, 0.32]. H3 was supported by a full mediation.

### 3.4. Mediation of Self-Compassion on Mental Health Shame—Mental Health Problems (H4)

Last, another path analysis was conducted, using Model 4 in the Process Macro (parallel mediation model [82]). Mental health shame was entered as a mediator variable, instead of negative mental health attitudes. Self-compassion (mediator variable) and mental health problems (outcome variable) remained the same.

Self-compassion partially mediated the relationship between mental health shame and mental health problems (Figure 2). The total effect of mental health shame on mental health problems, including self-compassion, was significant, *b* = 0.42, *t*(117) = 3.99, *p* = 0.0001, CI [0.21, 0.62]. The direct effect of mental health shame on mental health problems was also significant, *b* = 0.21, *t*(116) = 2.14, *p* = 0.03, CI [0.02, 0.40]. The indirect effect of mental health shame on mental health problems, controlling for self-compassion, was significant, *b* = 0.21, CI [0.09, 0.34]. H4 was supported by a partial mediation.

## 4. Discussion

Our aim was to explore how negative mental health attitudes, mental health shame, and self-compassion are associated with psychopathological symptoms. We hypothesised that all variables of interest would be significantly associated and predictive of mental health problems. While negative attitudes and mental health shame were significantly associated with mental health problems, they showed no significant predictive power for the mental health problems (H1ab, H2ab). On the other hand, self-compassion was both a significant correlate with, and predictor of mental health problems (H1c, H2c). Mediation analyses also indicated self-compassion fully mediated the relationship between negative mental health attitudes and mental health problems (H3), and partially mediated the relationship between mental health shame and mental health problems (H3, H4). Overall, the importance of self-compassion to mental health was demonstrated in all analyses.

Our findings support and add to the empirical literature of the beneficial relationship between practicing self-compassion and better mental health. Gilbert and Procter [31] argue that self-compassion can reduce feelings of embarrassment and humiliation, which provides further insight into the study findings. This is in line with earlier findings that self-compassion mediated the relationship between shame and negative body image [83]. Self-compassion has also been found to mediate the relationship between shame and symptoms of depression [84] and students’ psychological health [85]. This research supports the beneficial role of self-compassion on shame reduction [31,42,45,48]. Though mental health shame is often a by-product of socially constructed attitudes and beliefs about mental health [14], our findings indicate that despite their strong relationship in our sample, these negative attitudes do not directly affect mental health but somewhat compromise the positive impact of self-compassion on mental health.

Furthermore, in our sample, while negative mental health attitudes and mental health shame were significantly correlated with mental health problems, these did not significantly predict mental health problems. This stands in contrast to some earlier findings [43], but echoes others [33]. A plausible explanation for this could be that perceived attitudes and mental health shame are not precursors of mental health problems but rather co-occurrences and consequences. Therefore, mental health shame possibly only arises once mental health problems are experienced. Mental health shame, as measured in the current study, takes into account external, reflective, and internal shame of having mental health issues [36,86,87], hence it is possible that one’s own and the perceived attitudes of others towards mental health only become relevant when one becomes a potential target of these beliefs. There are some findings suggesting that anticipated stigma is significantly linked to higher rates of mental distress [88]. Indeed, this cross-sectional study did not appraise the causality, but our findings indicate a possible interpretation for non-predictive, yet correlative relationships of negative mental health attitudes and mental health shame with mental health problems. Further research is needed to investigate the directions of these relationships through longitudinal observation.

Applied to the context of mental health in the Czech Republic, our current findings indicate that the pervasive negative attitudes associated with mental health are strongly linked to feelings of shame, which have a direct effect on mental health. These attitudes and feelings of shame may create cognitive barriers for help-seeking behaviours that prevent individuals in distress from accessing mental health services and receiving treatment. It seems that the historical demonised image of psychiatry in former socialist societies [1] continues to cloud the cultural perception of mental health and therefore continues to fuel scepticism and shame [13]. This may be potentially aggravated and maintained by feelings of inferiority and elevated status anxiety in the country [89]. With this persistent fear of losing one’s standing, the threat system seems constantly activated, and with a culturally inclined tendency to restrain from pleasures and soothing behaviours [62,64], the emotion-regulation systems may be unbalanced. This imbalance could be a plausible explanation for the elevated prevalence of mental distress [71] in the Czech Republic [4], and addressing the same might be a feasible avenue to explore in the reform of mental health. Given that poor mental health is a nationally recognised problem, the importance of self-compassion needs to be emphasised in Czech higher education. Training that targets self-compassion would be helpful for the mental health of Czech students. Considering the heightened anxiety associated with the transition to university [90], the effects of such training may be maximised during the orientation/induction session.

### 4.1. Limitations

While the current study has added to the understanding of the relationship between negative mental health attitudes, mental health shame, self-compassion and mental health problems, it is not without limitations. One of the key limitations would be that our sample was recruited at one university in the Czech Republic. Moreover, we recruited psychology students, many of whom were female students. Research has shown that mental health literacy [91,92], as well as being female, is associated with less stigma and negative attitudes toward mental health [32]. These might have caused sample biases. Therefore, a more diverse sample of students needs to be involved. Second, self-report measures might compromise the accuracy of reporting because of response biases [93]. Moreover, process-oriented constructs such as self-compassion may require closer evaluation [94,95]. Third, the causal direction of these relationships has not been examined. Longitudinal studies would be helpful to elucidate the temporal patterning of the relationships identified in this study and may help develop more effective interventions.

### 4.2. Conclusions

In conclusion, this study highlights the relationship between greater self-compassion and better mental health in university students. Practicing self-compassion, particularly in student samples who experience high levels of stress and who may be at greater risk of developing mental health illness, may be beneficial. Future research should examine ways in which positive psychology techniques can be introduced into the university curriculum. Longitudinal research should also be conducted to examine the causal relationships between negative attitudes and shame towards mental health, self-compassion and poor mental health.

## Figures and Tables

**Figure 1 healthcare-10-00676-f001:**
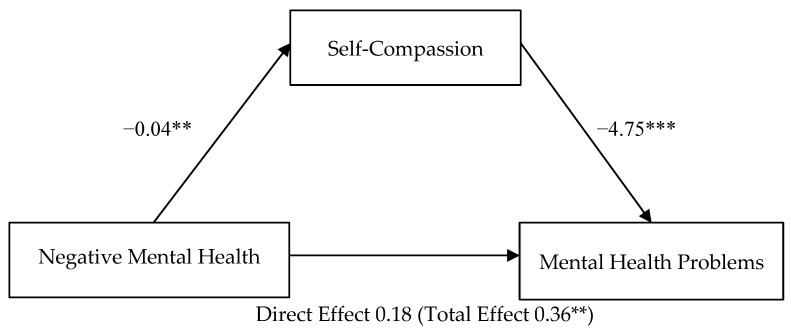
Parallel Mediation Model: Negative Mental Health Attitudes as a predictor of Mental Health Problems, mediated by Self-Compassion. Note: The confidence interval for the indirect effect is a BCa bootstrapped CI based on 5000 samples. Direct effect (total effect). Values attached to arrows are coefficients indicating impacts. ** *p* < 0.01, *** *p* < 0.001.

**Figure 2 healthcare-10-00676-f002:**
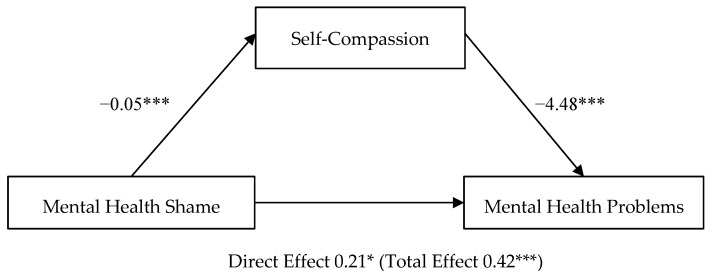
Parallel Mediation Model: Mental Health Shame as a predictor of Mental Health Problems, mediated by Self-Compassion. Note: The confidence interval for the indirect effect is a BCa bootstrapped CI based on 5000 samples. Direct effect (total effect). Values attached to arrows are coefficients indicating impacts. * *p* < 0.05, *** *p* < 0.001.

**Table 1 healthcare-10-00676-t001:** Descriptive statistics and correlations between mental health problems, negative mental health attitudes, mental health shame, and self-compassion in Czech psychology students (*n* = 119).

		M	SD	*α*	1	2	3	4	5	6
1	Gender (F = 93, M = 20, No Answer = 6)				-					
2	Age (19–44 in our sample)	21.87	3.32		0.25 **	-				
3	Mental Health Problems (0–126)	38.00	20.98	0.90	−0.14	0.02	-			
4	Negative Mental Health Attitudes (0–24)	4.50	4.63	0.86	0.20 *	0.08	0.26 **	-		
5	Mental Health Shame (0–81)	18.34	11.55	0.90	0.13	0.02	0.35 **	0.56 **	-	
6	Self-Compassion (1–5)	3.08	0.60	0.81	0.12	0.002	−0.54 **	−0.25 **	−0.36 **	-

Gender F = 0, M = 1; * *p* < 0.05, ** *p* < 0.01.

**Table 2 healthcare-10-00676-t002:** Multiple regression examining negative attitudes to mental health, shame, and self-compassion as predictors of mental health symptoms in Czech psychology students (*n* = 119).

	Dependent Variable: Mental Health Problems			
	B	SE_B_	β	95% CI
Step 1				
Gender	−0.62	0.43	−14	−1.46, 0.23
Age	0.02	0.06	0.03	−0.10, 0.14
Adj. R^2^	0.001			
Step 2				
Gender	−0.61	0.37	−14	−1.35, 0.13
Age	0.05	0.05	0.08	−0.05, 0.16
Negative Mental Health Attitudes	0.16	0.13	0.12	−0.10, 0.42
Mental Health Shame	0.14	0.12	0.12	−0.10, 0.38
Self-Compassion	−4.25	0.80	−0.46 ***	−5.84, −2.66
Adj. R^2^ Δ		0.29 ***		

Gender F = 0, M = 1; *** *p* < 0.001.

## Data Availability

The data presented in this study are available on request from the corresponding author. The data are not publicly available due to ethical restrictions.

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
