# Peer review of "Mental Health of Czech University Psychology Students: Negative Mental Health Attitudes, Mental Health Shame and Self-Compassion"

_healthcare, 2022, doi:10.3390/healthcare10040676_

Round 1
Reviewer 1 Report
Dear Authors,
Congratulations on a good article. It touches on extremely important mental health issues among young people, who are often left alone with their problems, often because of embarrassment.
The results you obtained are strongly (in +) related to the population you studied. If the respondents were other students, from other fields of study, surely the results would look different. So I think the title should be changed. It should be pointed out that the research was conducted among psychology students. Of course, you write this in the manuscript, but it is worth emphasizing it in the title in order not to mislead the potential reader.
It would be good to exclude from the discussion the limits and conclusions you reached. This will be a clearer record.
Similarly, I would suggest separating the following sections in the abstract: introduction, aim, material and method, results, conclusions.
In Linia 209 is wrong note [α=.86; 81].
Author Response
Response Letter
Manuscript ID: healthcare-1659170
"Mental Health of Czech University Students: Negative Mental Health
Attitudes, Mental Health Shame, and Self-Compassion”
Dear Reviewers,
Thank you for your helpful feedback. We have systematically revised our manuscript addressing the points you have raised. Please see our responses below. We hope this revised paper is now acceptable for publication. We extend our sincere gratitude to you for your feedback that has significantly helped to strengthen the paper.
Reviewer 1
Reviewer 1’s comment 1
Dear Authors,
Congratulations on a good article. It touches on extremely important mental health issues among young people, who are often left alone with their problems, often because of embarrassment.
The results you obtained are strongly (in +) related to the population you studied. If the respondents were other students, from other fields of study, surely the results would look different. So I think the title should be changed. It should be pointed out that the research was conducted among psychology students. Of course, you write this in the manuscript, but it is worth emphasizing it in the title in order not to mislead the potential reader.
Authors’ response 1-1
Thank you for your helpful and thoughtful feedback. In line with your comments, the title and abstract are revised clarifying those are psychology students.
Reviewer 1’s comment 2
It would be good to exclude from the discussion the limits and conclusions you reached. This will be a clearer record.
Similarly, I would suggest separating the following sections in the abstract: introduction, aim, material and method, results, conclusions.
Authors’ response 1-2
Thank you for your helpful feedback on the presentation. Now the discussion is revised accordingly. We were not able to change the abstract due to the requirement in the author instructions (i.e., one paragraph without headings); https://www.mdpi.com/journal/healthcare/instructions
Reviewer 1’s comment 3
In Linia 209 is wrong note [α=.86; 81].
Authors’ response 1-3
Thank you, now it is revised.
Reviewer 2 Report
Manuscript is well-written and highlights important and emerging research area in mental health problem with university students that significant around the world. I would like to suggest for Corresponding author response about manuscript review;
- Unfamiliar terms are not defined early in the topic paper.
- Check abstract result not show detail of Correlation; Regression and Path analysis. Please clarify & summarize here.
- Design - not show ID of IRB Ethic Review Board, Please put ID/ Number of your institution approve.
- The findings are interesting as intentions leads to performance which is a new dimension.
- Figure 2 - 3, Can you improve professional figure? Design quality of figure if possible to show professional style in manuscript.
- Suggestion: a much briefer and more direct article. There is good sample N selection. Group assignment should be clearer. The more complex analysis of variance should be used. And language help should be obtained for fit better.
- Several paragraphs require a clearer redaction style. Basic lexical and grammatical errors are also found in the manuscript.
- Please show your idea of "Limitation of the study"
Author Response
Response Letter
Manuscript ID: healthcare-1659170
"Mental Health of Czech University Students: Negative Mental Health
Attitudes, Mental Health Shame, and Self-Compassion”
Dear Reviewers,
Thank you for your helpful feedback. We have systematically revised our manuscript addressing the points you have raised. Please see our responses below. We hope this revised paper is now acceptable for publication. We extend our sincere gratitude to you for your feedback that has significantly helped to strengthen the paper.
Reviewer 2
Reviewer 2’s comment 1
Manuscript is well-written and highlights important and emerging research area in mental health problem with university students that significant around the world. I would like to suggest for Corresponding author response about manuscript review;
Unfamiliar terms are not defined early in the topic paper.
Authors’ response 2-1
Thank you for your thoughtful comment. In line with your suggestion, now relevant terms are defined.
Reviewer 2’s comment 2
Check abstract result not show detail of Correlation; Regression and Path analysis. Please clarify & summarize here.
Authors’ response 2-2
The results of those analyses were outlined under that sentence.
Reviewer 2’s comment 3
Design - not show ID of IRB Ethic Review Board, Please put ID/ Number of your institution approve.
Authors’ response 2-3
Now these details are provided.
Reviewer 2’s comment 4
The findings are interesting as intentions leads to performance which is a new dimension.
Figure 2 - 3, Can you improve professional figure? Design quality of figure if possible to show professional style in manuscript.
Authors’ response 2-4
In line with your comment, the figures are refined, while still following the author instructions.
Reviewer 2’s comment 5
Suggestion: a much briefer and more direct article. There is good sample N selection. Group assignment should be clearer. The more complex analysis of variance should be used. And language help should be obtained for fit better. Several paragraphs require a clearer redaction style. Basic lexical and grammatical errors are also found in the manuscript.
Authors’ response 2-5
In line with your comment, more concise writing and/or clear writing has been applied where appropriate. Proofread has been carefully done by all authors.
Reviewer 2’s comment 6
Please show your idea of "Limitation of the study".
Authors’ response 2-6
In line with your comment, now the limitation section is clearly placed.
Thank you for your helpful feedback.